# Diffusion Beats ARM: Diffusion Large Language Models for Generative Recommendation

## Abstract

As a promising new paradigm, generative recommender systems frame recommendation as a process of learning data distributions, enabling content creation and diversity exploration by modeling patterns in user behaviors or item characteristics. A common practice to handle large-scale item catalogs is to quantize different item features into discrete semantic sequences, which are then used to train large language models for item generation. However, we argue that this autoregressive generation approach is fundamentally misaligned with the nature of item features in recommendation. Unlike natural language, item attributes are parallel, intertwined, and mutually defining—lacking the hierarchical and sequential dependency that autoregressive models assume. This misalignment limits the effectiveness of existing generative recommendation methods. To address this issue, we propose a new generative recommendation paradigm called GREED (**G**enerative **RE**commendation via **E**lemental **D**iffusion over Large Language Models). Instead of relying on sequential generation, GREED leverages diffusion-based generative modeling to capture the joint distribution of item features in a non-autoregressive manner. This design better respects the parallel structure of item attributes, thereby improving both efficiency and ranking performance. Extensive experiments demonstrate that GREED outperforms state-of-the-art methods on multiple benchmark datasets. We also conduct detailed offline analyses to validate the efficiency and effectiveness of our approach.

## 1 Introduction

As a cornerstone technology underpinning modern digital platforms, recommendation systems (Resnick & Varian, 1997; Bobadilla et al., 2013; Isinkaye et al., 2015) have achieved widespread adoption in industrial applications. These systems utilize users' historical behaviors and interest profiles to identify potentially relevant items, thereby improving user engagement and satisfaction (Cheng et al., 2016; Covington et al., 2016; Geng et al., 2022; Gomez-Uribe & Hunt, 2015). The architecture of industrial recommendation systems generally follows a multi-stage funnel-shaped workflow. Starting from a large-scale item corpus, the system progressively narrows down candidate items through the coordinated operation of multiple specialized models. The pipeline consists of two main stages: retrieval and ranking. During the retrieval phase, lightweight models efficiently scan vast item repositories, often comprising millions of entries, to produce a manageable candidate set that balances relevance and computational efficiency. In the subsequent ranking stage (Karatzoglou et al., 2013), more sophisticated models are employed to accurately assess and order these candidates based on predicted user preference scores. This hierarchical design enables platforms to optimize both recommendation quality and system performance, ultimately delivering personalized item rankings that maximize relevance and user satisfaction.

Of the various discriminative approaches in recommendation systems, several effective and standardized methods have emerged. Notably, a range of techniques incorporate sequence modeling to capture user session dynamics for subsequent item recommendation (Hidasi et al., 2015; Li et al., 2017; Sun et al., 2019; Wu et al., 2019; Zhang et al., 2019). Building upon advances in large language models (LLMs) (Brown et al., 2020), generative recommendation systems (GRs) have garnered significant research attention. To adapt generative models for item recommendation, one line

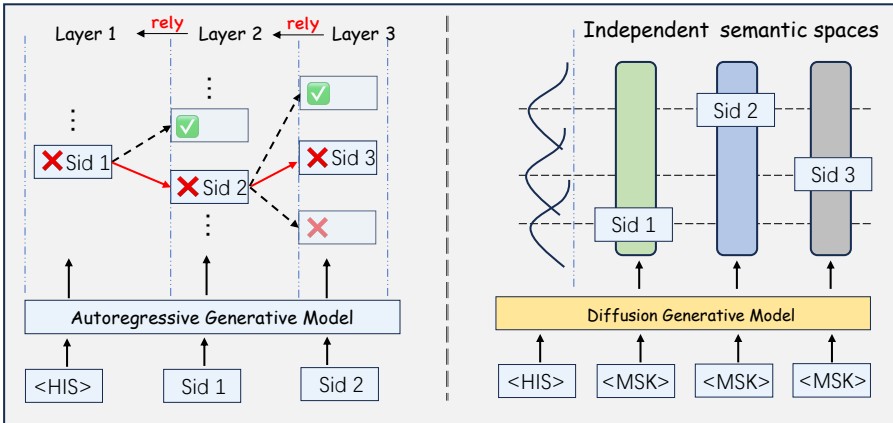

Figure 1: A Comparative Analysis of ARMs versus DiffusionLLM-Based Paradigms for GRs.

of work decomposes items into discrete or numerical features and incorporates them directly into the language model's vocabulary (Zhai et al., 2024). While this approach can, in theory, harness the benefits of scaling laws (Kaplan et al., 2020; Zhang et al., 2024), it often necessitates an impractically large vocabulary, leading to substantial computational overhead. Another prevalent strategy, which we refer to as Discrete Quantized Generative Recommendation (DQGRs), represents items as discrete sequences derived from quantized item embeddings (Rajput et al., 2023; Wang et al., 2024; Zheng et al., 2024; Deng et al., 2025). The effectiveness of this paradigm hinges critically on the quality of the discrete item representations. Current methods primarily rely on techniques such as vector quantization (VQ) (Zheng et al., 2024) or residual quantization (RQ) (Lee et al., 2022), which often face limitations in representation capacity and semantic fidelity.

*What constitutes an effective representation and generation paradigm for DQGRs?* Addressing this question requires an understanding of the core demands of recommendation systems. In practice, a recommendation—such as a shirt selected based on its *color*, *brand*, and *style*—involves multifaceted attributes that are inherently independent. Prevailing generative recommendation over Autoregressive Models (ARMs), however, formulate item generation as a rigid, sequential prediction of tokens. This approach imposes an artificial chain of dependencies among attributes (e.g., predicting the brand only after the color), which misrepresents their parallel nature. Moreover, the causal attention mechanism in ARMs leads to an error accumulation effect, where a mistake in generating one feature propagates through the subsequent generation steps, compromising the integrity of the final item representation. which is shown in Figure 1. These observations lead us to propose that an ideal DQGR framework should fulfill four key criteria:

- **Multi-dimensional Representation.** The model should treat different item attributes (*e.g.*, category, brand) as equally important dimensions, rather than imposing a sequential hierarchy.

- **High Representation Space Utilization.** Given the vast and growing item catalogs in practice, the discrete encoding must make efficient use of its vocabulary to avoid codebook collapse and represent a wide variety of items uniquely.

- **Preservation of Semantic Topology.** Items that are similar in the continuous embedding space (*e.g.*, two action movies by the same director) should remain close in the discrete representation space to reflect user behavioral patterns.

- **Global Context Awareness.** The generation of an item should be decided holistically, considering all user interactions and attributes simultaneously, rather than being constrained by a local, token-by-token autoregressive process.

Nevertheless, existing DQGR paradigms continue to exhibit notable shortcomings. Approaches relying on VQ are susceptible to severe codebook collapse, thereby compromising the objective of high representation space utilization—a particularly critical requirement given the scale of real-world item catalogs. In contrast, RQ imposes artificial sequential dependencies among originally orthogonal feature dimensions, contradicting the principle of multi-dimensional representation. The

ARM architecture, reliant on left-to-right causal attention, is ill-suited for item generation due to the absence of a canonical feature order. This inherent misalignment impedes global context integration and results in errors that propagate in an autoregressive manner. To address these challenges, we propose **G**enerative **RE**commendation via **D**iscrete **E**lemental Indexing over Diffusion Large Language Models named GREED. Recognizing that the performance of generative recommendation models depends critically on both the scale of items and the quality of discrete encoding, we introduce a new quantization technique called uniform implicit quantization UIQ, which effectively encodes large item sets into discrete code sequences while enriching the multidimensional representation of items and mitigating codebook collapse. In addition, to overcome the limitation of the item representation ability of the left-to-right generation in existing GRs methods, we adopt a discrete diffusion generative paradigm. In contrast to conventional ARM-based next-token prediction, our model supports parallel multi-token prediction, alleviating efficiency bottlenecks and enabling the modeling of longer sequences. We conduct extensive experiments on multiple recommendation benchmarks, demonstrating that our approach consistently outperforms state-of-the-art baselines. Ablation studies and theoretical analysis further validate the efficacy of the proposed design. The main contributions of this work are summarized as follows:

- This paper introduces a novel Uniform Implicit Quantization method for learning discrete representations of items. Furthermore, we provide a theoretical analysis demonstrating that UIQ achieves greater codebook utilization, thereby alleviating the codebook collapse issue.

- To address the limitations of the left-to-right ARMs paradigm in capturing complex item semantics, we introduce a novel discrete diffusion model for multi-token generative recommendation. Our method plans the generation process from a global perspective, making it more suited to modeling the rich semantic information of items in recommendation scenarios.

- Empirical results show that GREED achieves state-of-the-art performance across multiple datasets and evaluation metrics. A key advantage of our approach is its ability to precisely balance performance and efficiency by varying the number of tokens generated per step.

## 2 PRELIMINARY

In this section, we present the preliminary concepts and definitions that are essential for understanding the subsequent discussions in this paper.

### 2.1 GENERATIVE RECOMMENDER SYSTEMS

We formalize the generative recommendation task as follows. Let $\mathcal{X}$ denote the entire item space, where an item $\Phi \in \mathcal{X}$ is represented as a structured token sequence $C = (c_0, ..., c_j) = \mathcal{E}(\Phi)$ via an encoding function $\mathcal{E}$. Each token $c_i$ may correspond to a discrete feature or a semantic identifier. Given a user, represented by a feature vector $u \in \mathcal{U}$ and an interaction history $\mathcal{H} = (\Phi_1^h, \ldots, \Phi_M^h)$, the objective of a generative model $\mathcal{G}$ is to train a generative model $\mathcal{G}$ such that the conditional distribution it models, $P(C \mid u, \mathcal{H})$, approximates the true target distribution $P(C_{\text{target}})$. This is achieved by minimizing the Kullback-Leibler (KL) divergence between them:

$$\min_{\mathcal{G}} D_{\text{KL}} \left[ P(C_{\text{target}}) \mid P(C \mid u, \mathcal{H}) \right] \tag{1}$$

### 2.2 DISCRETE DIFFUSION LANGUAGE MODELS

Owing to their stable generation process and high output quality, diffusion models are now effectively applied to discrete data generation. Specifically, discrete diffusion language models achieve this by defining a Markov chain directly on discrete spaces, thus facilitating the generation of sequential data like natural language. In our work, adhering to the formalism in (Nie et al., 2025; Sahoo et al., 2024), we represent discrete variables of $K$ categories as one-hot vectors and define the categorical distribution as $\text{Cat}(\cdot; \pi)$, where $\pi$ is the probability vector over the categories.

**Forward Masking Process** The discrete masked diffusion language model defines a forward noising process $q$ that transforms clean data $\mathbf{x}$ into a sequence of increasingly noisy latent variables $\mathbf{z}_t$ along a continuous time index $t$. At $t = 0$, the latent variable equals the original data ($\mathbf{z}_0 = \mathbf{x}$), and at the terminal time $t = 1$, the data is fully corrupted, meaning all tokens are replaced by a designated

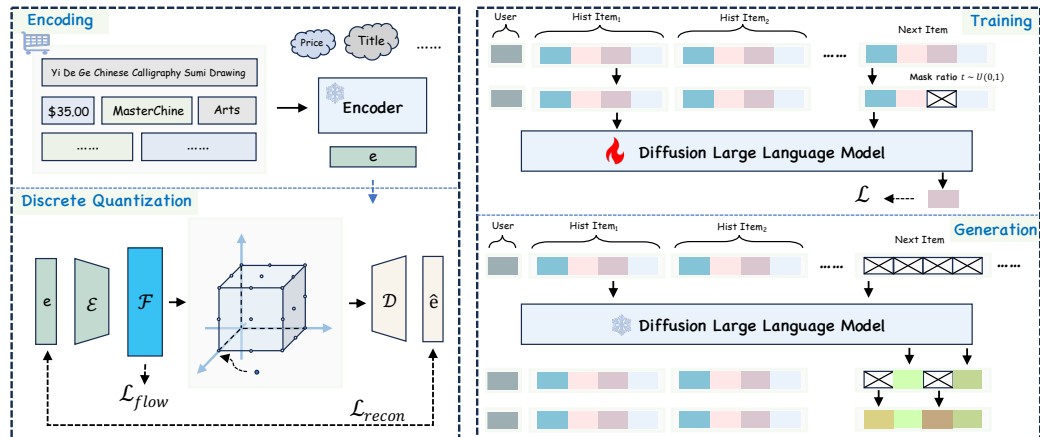

Figure 2: **The framework of our** GREED. We translate item content to discrete sequences which can be acted as generative objectives for generation models (left). With discrete semantic ids, we design a multi-token diffusion language model to execute item generation task (right). a) For encoding stage, we use a freezed pre-trained model to obtain the representations of items. b) For quantization stage, $\mathcal{E}$, $\mathcal{F}$ and $\mathcal{D}$ represent encoder, flowing model, and decoder, respectively. And $e$ denotes representations of items from encoding stage. The discrete scalar quantization will quantizate the item represntations into a hypercube, we use cubes to represent the explanation for simplicity. c) For generation model partition, we design a series of generation tasks to supervise model (*e.g*, multi-token generation task, next-item generation task).

[MASK] token $\mathbf{m}$. This process is governed by a Markov chain (Norris, 1998). The marginal probability distribution of the latent variable $\mathbf{z}_t$ at any time $t$, given the initial data $\mathbf{x}$, is given by the following categorical distribution:

$$q(\mathbf{z}_t|\mathbf{x}) = \text{Cat}(\mathbf{z}_t; \alpha_t\mathbf{x} + (1-\alpha_t)\mathbf{m}) \tag{2}$$

Here, $\alpha_t \in [0,1]$ is a strictly decreasing function of $t$ satisfying $\alpha_0 \approx 1$, $\alpha_1 \approx 0$, which controls the probability of a token remaining unchanged. During the forward diffusion process, this scheduling function progressively shifts probability mass from the original tokens toward the [MASK] token.

**Reverse Unmasking Process** The objective of the reverse process $p_\theta$ is to learn how to reverse the forward diffusion process, thereby recovering the original clean data $\mathbf{x}$ from the masked latent $\mathbf{z}_t$. This process is defined by the marginal distribution $p_\theta(\mathbf{x}) = \int_{\mathbf{z}} p_\theta(\mathbf{z}_1)p_\theta(\mathbf{x}|\mathbf{z}_0)\prod_{i=1}^{T}p_\theta(\mathbf{z}_s|\mathbf{z}_t)d\mathbf{z}_{0:1}$, which is approximated through a neural network $\mathbf{x}_\theta(\mathbf{z}_t, t)$. Using a substitution-based parameterization, the definition of $p_\theta(\mathbf{z}_s|\mathbf{z}_t)$ is as follows.

$$p_\theta(\mathbf{z}_s|\mathbf{z}_t) = \begin{cases} \text{Cat}(\mathbf{z}_s; \mathbf{z}_t) & \mathbf{z}_t \neq \mathbf{m}, \\ \text{Cat}\left(\mathbf{z}_s; \dfrac{(1-\alpha_s)\mathbf{m} + (\alpha_s - \alpha_t)\mathbf{x}_\theta(\mathbf{z}_t, t)}{1-\alpha_t}\right) & \mathbf{z}_t = \mathbf{m}. \end{cases} \tag{3}$$

## 3 METHODS

Our proposed framework comprises two main stages: a quantization stage and a generation stage, as illustrated in Figure 2. During the quantization stage, items are represented as discrete semantic ID sequences. To achieve this discrete quantization, we first utilize a pre-trained language model (*e.g*, T5 (Raffel et al., 2020) or BERT (Devlin et al., 2019)) to obtain item representations. Subsequently, we introduce a novel discrete diffusion generative paradigm for recommendation.

### 3.1 DISCRETE QUANTIZATION BOTTLENECK

To leverage generative models (*e.g*, LLMs) for recommendation tasks, abundant item features must be transformed into discrete tokens or semantic IDs. This transformation significantly compresses

the generation length required for items and enables LLMs to model item distributions. Crucially, the quality of the discrete representation directly impacts retrieval and ranking performance in the subsequent generation stage. VQ employs learnable codebooks (defined by the number of codebooks, $B$, and codebook size, $K$) to quantize data. Each code in the codebook is a learnable embedding vector designed to aggregate common data features. However, optimization objectives can lead to codebook collapse, where the model converges on using only $n$ features to minimize quantization loss. In severe cases, $n$ can be significantly smaller than $K(n \ll K)$, resulting in many codes remaining inactive with extremely low utilization rates. RQ employs multiple parallel, hierarchically structured codebooks. The causal dependencies between quantization layers introduce significant complexity during both training and inference. Within recommendation systems, where items possess rich, multi-dimensional feature information, preserving these multi-dimensional semantic relationships during quantization is essential. Both the codebook collapse inherent in VQ and the causal chain dependencies introduced by RQ impede the effective preservation of rich item features. This limitation poses significant challenges for downstream generation tasks.

Inspired by Mentzer et al. (2023); Yu et al. (2023), we explore discrete quantization of items using scalars. For a $d$-dimensional item representation $\mathbf{z} \in \mathbb{R}^d$, our objective is to quantize $\mathbf{z}$ into a finite set of uniformly spaced codewords. By applying a bounding function $f$ (*e.g.* $f : \mathbf{z} \mapsto \lfloor \frac{L}{2} \rfloor \mathbf{tanh}(\mathbf{z})$), the features can be quantized to one of $L$ discrete values $\mathbf{q} = \mathbf{round}(f(\mathbf{z}))$, where $\mathbf{q} \in \mathcal{C}$. Here, $\mathcal{C}$ serves as an implicit codebook with a capacity of $|\mathcal{C}| = L^d$. However, in recommendation systems, the distribution of item representations often exhibits high concentration due to data sparsity and feature specificity. To illustrate this phenomenon, consider a component $z_i$ of $\mathbf{z}$ following a Gaussian distribution with mean 0 and variance $\sigma^2$:

$$p(z_i) = \frac{1}{\sqrt{2\pi}\sigma} e^{-z_i^2/(2\sigma^2)} \tag{4}$$

The distribution of $u_i = \tanh(z_i)$ then deviates from Gaussian, becoming concentrated around zero with sparse tails. Specifically, the probability density function of $u_i$ is:

$$p(u_i) = p(z_i)|\frac{dz_i}{du_i}| = \frac{1}{\sqrt{2\pi}\sigma} e^{-[\mathrm{atanh}(u_i)]^2/(2\sigma^2)} \cdot \frac{1}{1-u_i^2} \tag{5}$$

This density peaks at $u_i \approx 0$ and approaches zero as $u_i \to \pm 1$. Consequently, the values of $f(\mathbf{z})$ are naturally concentrated, which contradicts the desired uniform distribution across quantization levels for optimal codebook utilization. This distribution mismatch makes it difficult for the encoder, guided solely by the reconstruction loss, to effectively learn a representation that leverages the full capacity of the codebook, potentially leading to suboptimal convergence. To address this fundamental issue, we propose uniform implicit quantization UIQ. The core idea is to explicitly transform the latent distribution before quantization to minimize expected quantization distortion and maximize codebook utilization. The theoretical superiority of our UIQ approach is grounded in the following corollary, which establishes the optimality of uniform input distributions for finite scalar quantization, the details of the proof can be found in Appendix B.1:

**Corollary 1** (**Optimality of Uniform Input Distribution for Scalar Quantization**) *Let $\mathbf{z} \in \mathbb{R}^d$ be a continuous latent representation, and let $\mathbf{q} = \mathbf{round}\left[f(\mathbf{z})\right] \in \{0, 1, \ldots, L-1\}^d$ denote its finite scalar quantization with $L$ levels per dimension. If the input $\mathbf{z}$ follows a standard uniform distribution with independent dimensions, i.e., $\mathbf{z} \sim U(0, 1)^d$, then:*

- *The expected quantization distortion $\mathbb{E}[\mathcal{D}_{\mathrm{quant}}]$ is minimized, which ensures maximal preservation of original information in the discrete representation.*

- *The entropy of the discrete code $\mathbf{q}$ is maximized, reaching the upper bound of $d \cdot \log L$, which guarantees optimal codebook utilization and expressive power.*

Our approach is grounded in the Probability Integral Transform (PIT) (Angus, 1994), which states that for a continuous random variable $X$ with CDF $F_X$, the variable $Y = F_X(X)$ follows a standard uniform distribution, $Y \sim U(0, 1)$. This provides a principled way to achieve our goal: if we can apply the CDF of $z_i$ to itself, the result will be uniformly distributed.

Since the true CDF of $z_i$ is typically unknown, we approximate it using a learnable function $g(z; \theta)$ parameterized by a normalizing flow (NF) (Papamakarios et al., 2021; Dinh et al., 2016; Huang

et al., 2018; Cao et al., 2019). Normalizing flows are ideal for this task as they can learn complex, invertible transformations and exact probability densities. We train the flow model by maximizing the likelihood of the data, which is equivalent to minimizing the negative log-likelihood (NLL) loss:

$$\mathcal{L}_{\textbf{flow}} = -\mathbb{E}_z \Big[ \log p_Y(g_\theta(z)) + \log |\textbf{det} J_{g_\theta}(z)| \Big] \tag{6}$$

$$= -\frac{1}{N} \sum_{i=1}^{N} \Big[ \log |\textbf{det} J_{g_\theta}(z)| \Big] \tag{7}$$

Then, we apply a bounding function $f$ ($f : \mathbf{z} \mapsto \textbf{round}\Big[(L-1)\textbf{sigmoid}(\mathbf{z})\Big]$) to quantify items. In this way, we can quantify the input to scalar points of $L$ levels. Through reconstructing item representations as optimum objective, we use reconstruction loss as follows.

$$\mathcal{L}_{\textbf{recon}} = \frac{1}{N} \sum_{i=1}^{N} (e_i - \hat{e}_i)^2 \tag{8}$$

In the end, the total quantization optimum loss is as follows.

$$\mathcal{L}_{\textbf{quan}} = \lambda \mathcal{L}_{\textbf{recon}} + (1-\lambda) \mathcal{L}_{\textbf{flow}} \tag{9}$$

### 3.2 DISCRETE DIFFUSION GENERATION RECOMMENDER ARCHITECTURE

To overcome the limitations of capturing complex multi-granularity features and global planning ability in current generative recommendation, we introduce a new paradigm based on the diffusion language model. Unlike natural language processing, where tokens often contribute incrementally to semantics, in recommendation systems, the entire generated token sequence collectively defines a single item. ARMs, due to their strict left-to-right, token-by-token generation process, struggle to capture the global semantic representation of an item during decoding. In contrast, diffusion language models exhibit stronger global decision-making and planning abilities through iterative refinement across the entire sequence, as shown in recent studies (Kim et al., 2025; Ye et al., 2024).

Considering that an item is represented by a discrete sequence of ids, we group id tokens of generative response into item blocks of length $d$ (i.e. the length of discrete sequence of item) (Arriola et al., 2025; Huang & Tang, 2025). For a sequence consisting of $N$ items, each item sequence $C = (c_0, ..., c_d)$ can be represents a token block $\mathbf{x}^{(n-1) \cdot d : n \cdot d}$, which we simplify it as $\mathbf{x}^n$ for block $n \in \{1, ..., N\}$. Furthermore, in order for the model to output list-wise results, that is, to provide the next sequential recommendation result $item_{i+1}$ based on the item results $item_i$ generated by the model, we perform autoregressive decomposition of the model's likelihood at the block level, and its log-likelihood is as follows.

$$\log p_\theta(\mathbf{x}) = \sum_{n=1}^{N} \log p_\theta(\mathbf{x}^n | \mathbf{x}^{<n}) \tag{10}$$

where conditional probability $p_\theta(\mathbf{x}^n | \mathbf{x}^{<n})$ of item block $n$ is modeled by discrete masked denoising diffusion model. Specifically, a reverse diffusion process is defined by marginalizing the latent variables. In this process, the state at an earlier time step $s$ (where $s < t$) can be inferred from the state at $t$.

$$p_\theta(\mathbf{x}_s^n | \mathbf{x}_t^n, \mathbf{x}^{<n}) = \sum_{\mathbf{x}^n} q(\mathbf{x}_s^n | \mathbf{x}_t^n, \mathbf{x}^n) p_\theta(\mathbf{x}^n | \mathbf{x}_t^n, \mathbf{x}^{<n}) \tag{11}$$

where $q(\mathbf{x}_s^n | \mathbf{x}_t^n)$ is the conditional probability in the forward diffusion process. When generating each item block, the model iteratively recovers a clean item id sequence representation from the noise through a diffusion process, and this recovery process conditionally depends on all the previous item blocks that have been generated. Based on such a design, we generate candidate items in an item block through a discrete denoising process, which is a multi-token prediction process, thereby achieving an accelerated process. Moreover, this process model is based on the disordered generation of global planning, and the model can better capture the complex relationships within the discrete sequence of ids of items $C$. After the generation of an item block is completed, the entire generation sequence will be generated in an autoregressive manner in the order between the item blocks, thereby achieving list-wise sequential generative recommendation.

---

**Algorithm 1:** GREED Generation Stage

---

**Input:** $N$: The number of item block, $d$: The length of item block, $\mathbf{x}_\theta$: Denoising neural network, $T$: The time steps of diffusion denoising, Sampling_function: Reverse diffusion sampling function, $\mathbf{m}$: The special [MASK] token

**Output:** $X$: The generated item sequence

---

$X \leftarrow \emptyset$                        **Function:** Sampling()

**For** $n \leftarrow 1$ **to** $N$ **do**         **Input:** $\mathbf{x}_n^{t_j}$, logits$_{\mathbf{x}_n}$, $t_j$, $t_{j-1}$

    $X^{<n} \leftarrow X$                    **Output:** $\mathbf{x}_n^{t_{j-1}}$

    $\mathbf{x}_n^{t_T} \leftarrow [\mathbf{m}] \times d$

    **For** $j \leftarrow T$ **downto** $1$ **do**     **For** $l \leftarrow 1$ **to** $d$ **do**

        $t_j \leftarrow j/T$                $P(\mathbf{x}_n^{t_{j-1},l}|\mathbf{x}_n^{t_j,l}, \hat{x}_n^l) = q(\mathbf{x}_n^{t_{j-1},l}|\mathbf{x}_n^{t_j,l}, \hat{x}_n^l)$

        $t_{j-1} \leftarrow (j-1)/T$        // Sample token at position $l$

        logits$_{\mathbf{x}_n} \leftarrow \mathbf{x}_\theta(\mathbf{x}_n^{t_T}, X^{<n})$     $\mathbf{x}_n^{t_{j-1}} \sim P(\mathbf{x}_n^{t_{j-1},l}|\mathbf{x}_n^{t_j,l}, \hat{x}_n^l)$

        $\mathbf{x}_n^{t_{j-1}} \leftarrow$          $\mathbf{x}_n^{t_{j-1}} = (x_n^{t_{j-1},1}, ..., x_n^{t_{j-1},d})$

        Sampling$(\mathbf{x}_n^{t_{j-1}}, $logits$_{\mathbf{x}_n}, t_j, t_{j-1})$    **Return** $\mathbf{x}_n^{t_{j-1}}$

    $\mathbf{x}_n \leftarrow \mathbf{x}_n^{t_0}$

    $X \leftarrow X \oplus \mathbf{x}_n$

**Return** $X$

---

### 3.3 TRAINING AND GENERATION STAGE

The model is trained by minimizing a variational objective, specifically the Negative Evidence Lower Bound (NELBO). The NELBO provides a tractable upper bound on the true negative log-likelihood of the data. This objective is designed to facilitate efficient learning of the block-wise denoising process while capitalizing on the autoregressive dependencies between consecutive blocks. The overall training objective is formulated as follows:

$$\mathcal{L}(\mathbf{x}; \theta) = \sum_{n=1}^{N} \int_0^1 \left[ \sum_{\mathbf{z} \in \mathcal{V}^d} q(\mathbf{x}_t^n = \mathbf{z}|\mathbf{x}^n) \cdot \left( \frac{\alpha_t'}{1-\alpha_t} \log p_\theta(\mathbf{x}^n|\mathbf{x}_t^n = \mathbf{z}, \mathbf{x}^{<n}) \right) \right] dt \quad (12)$$

where $\alpha_t$ is the noise schedule function, a function defined on the interval $[0, 1]$, satisfying $\alpha_0 = 1$ and $\alpha_1 = 0$, and strictly decreasing with respect to $t$. During the training process, if the mask rate is too high or too low, it will lead to large variance of model training gradient and poor learning signal. In order to avoid the poor quality of model training due to the high gradient variance of the diffusion target during training, following the experience of the (Arriola et al., 2025), we clip the noise schedule through setting $\beta$ and $\gamma$, which can make $1 - \alpha_t \sim U[\gamma, \beta + \gamma]$.

The generation stage produces recommendations in a block-wise manner to ensure output coherence and computational efficiency. The process begins with a fully masked sequence of item semantic ids. Within each block, generation proceeds through multiple diffusion denoising steps, each refining a subset of tokens. Once a block is fully generated, it is unveiled to condition the autoregressive generation of the next block, thereby progressively constructing the complete sequence. The formal algorithm is detailed in Algorithm 1.

## 4 EXPERIMENTS

**Datasets.** We evaluated the proposed method on several widely adopted public benchmarks derived from the Amazon Product Reviews dataset (2018) (He & McAuley, 2016), which contains user reviews of products on Amazon from May 1996 to October 2018. Specifically, we used three product categories for the recommendation task: Arts, Video Games and Music Instruments. Detailed statistics and preprocessing procedures for these datasets are provided in Appendix C.

**Evaluation Metrics.** We use Normalized Discounted Cumulative Gain (NDCG@K) and top-$k$ Recall (Recall@K) with $K = \{1, 5, 10\}$ to evaluate the recommendation performance.

**Baselines.** We compared our proposed generative recommendation approach named GREED with the following sequential recommendation baselines: 1) Discriminate recommendation methods:

Table 1: Performance of our GREED with other methods on recommendation. Among them, those marked with the † symbol are the methods for discriminative recommendation methods.

| Datasets | Metrics | GRU4Rec† | HGN† | BERT4Rec† | SASRec† | S³Rec† | P5 | TIGER | OneRec | GREED [ours] | Improv. |
|---|---|---|---|---|---|---|---|---|---|---|---|
| **Arts Crafts Sewing** | Recall@1 | 0.0226 | 0.0248 | 0.0250 | 0.0235 | 0.0249 | 0.0249 | 0.0329 | 0.0357 | **0.0410** | +14.85% |
| | Recall@5 | 0.0344 | 0.0329 | 0.0355 | 0.0357 | 0.0371 | 0.0321 | 0.0363 | 0.0498 | **0.0520** | +4.42% |
| | NDCG@5 | 0.0285 | 0.0289 | 0.0308 | 0.0294 | 0.0308 | 0.0263 | 0.0346 | 0.0428 | **0.0464** | +8.41% |
| | Recall@10 | 0.0402 | 0.0393 | 0.0438 | 0.0389 | 0.0456 | 0.0344 | 0.0456 | 0.0544 | **0.0568** | +4.41% |
| | NDCG@10 | 0.0284 | 0.0307 | 0.0330 | 0.0305 | 0.0336 | 0.0277 | 0.0375 | 0.0443 | **0.0479** | +8.13% |
| **Video Games** | Recall@1 | 0.0070 | 0.0060 | 0.0073 | 0.0085 | 0.0113 | 0.0108 | 0.0080 | 0.015 | **0.0194** | +29.33% |
| | Recall@5 | 0.0212 | 0.0239 | 0.0244 | 0.0211 | 0.0359 | 0.0138 | 0.0395 | 0.0402 | **0.0449** | +11.69% |
| | NDCG@5 | 0.0140 | 0.0176 | 0.0157 | 0.0148 | 0.0237 | 0.0121 | 0.0263 | 0.0269 | **0.0336** | +24.91% |
| | Recall@10 | 0.0361 | 0.0386 | 0.0407 | 0.0340 | 0.0575 | 0.0154 | 0.0464 | 0.0563 | **0.0613** | +6.61% |
| | NDCG@10 | 0.0188 | 0.0230 | 0.0209 | 0.0190 | 0.0307 | 0.0126 | 0.0279 | 0.0321 | **0.0362** | +12.77% |
| **Music Instrument** | Recall@1 | 0.0146 | 0.0147 | 0.0187 | 0.0148 | 0.0198 | 0.0285 | 0.0056 | 0.0306 | **0.0340** | +11.11% |
| | Recall@5 | 0.0289 | 0.0263 | 0.0296 | 0.0296 | 0.0389 | 0.0350 | 0.0393 | 0.0384 | **0.0487** | +23.92% |
| | NDCG@5 | 0.0219 | 0.0246 | 0.0241 | 0.027 | 0.0293 | 0.0317 | 0.0331 | 0.0344 | **0.0420** | +22.09% |
| | Recall@10 | 0.0371 | 0.0390 | 0.0385 | 0.0371 | 0.0474 | 0.0464 | 0.0449 | 0.0509 | **0.0533** | +4.72% |
| | NDCG@10 | 0.0267 | 0.0293 | 0.0269 | 0.0294 | 0.0250 | 0.0353 | 0.0333 | 0.0384 | **0.0436** | +13.54% |

GRU4Rec (Hidasi et al., 2015), Bert4Rec (Sun et al., 2019), HGN (Ma et al., 2019), SASRec (Kang & McAuley, 2018), S³Rec (Zhou et al., 2020); 2) Generative recommendation methods: P5 (Geng et al., 2022), TIGER (Rajput et al., 2023), OneRec (Deng et al., 2025). To ensure fairness in comparison, we followed the reproduction schemes for different baseline methods in TIGER (Rajput et al., 2023). A detailed description of all baseline methods can be found in Appendix D.

## 4.1 OVERALL PERFORMANCE

In this section, we compared our approach with other sequential recommendation methods. The details of performance are shown in Table 1. Based on there results, we can find:

Among discriminative methods, S³-Rec achieves top performance. Its success stems from the bidirectional Transformer architecture, which comprehensively models user interaction sequences to capture complex dependencies and improve context awareness. This capability validates the core insight of our work. Furthermore, its pre-training strategy learns rich representations from massive data, effectively mitigating data sparsity and demonstrating robustness on long-tail items, making it the strongest discriminative baseline.

Of the generative methods evaluated, P5 demonstrates the weakest performance. This is owing to its use of atomic IDs for users and items, which lack semantic meaning and thus limit generalization capability, particularly resulting in a cold-start problem for new items. Our proposed GREED framework consistently achieves state-of-the-art results across all benchmarks. On the Arts dataset, GREED improves NDCG@5 by 8.41% and Recall@5 by 4.42% over the second-best baseline (OneRec). Further, it surpasses competing methods by 4.41% in Recall@10 and 8.13% in NDCG@10. Significant gains are also observed on the Video Games and Musical Instruments datasets. Notably, GREED yields remarkable improvements in Recall@1—by 14.85%, 42.67%, and 11.11% on the Arts, Video, and Music datasets, respectively. These gains can be attributed to GREED's use of a diffusion language model with a bidirectional attention mechanism, which captures multi-scale item features and supports global planning during generation. By first predicting the most confident semantic identifier and iteratively refining the sequence using full-context attention, GREED produces higher-quality item representations, leading to substantially improved recommendation accuracy.

## 4.2 NEW ABILITIES

In this section, we describe multiple new capabilities that directly follow from our proposed new generative recommendation paradigm, including conditional item recommendation task based on specified feature dimensions, recommendation ranking task, and continual recommendation.

**Conditional Item Recommendation** Unlike previous ARM architecture solutions, the new recommendation generative model paradigm we propose can handle any completion task by discarding

the causal attention mechanism. This endows our model with the ability to recommend conditional items (Iqbal et al., 2019). By post-processing the discrete representation of the obtained item, we can obtain the item features corresponding to different semantic ids, and thereby achieve the conditional item recommendation task that meets the user's requirements. The experimental results can be found in the Appendix F.3.

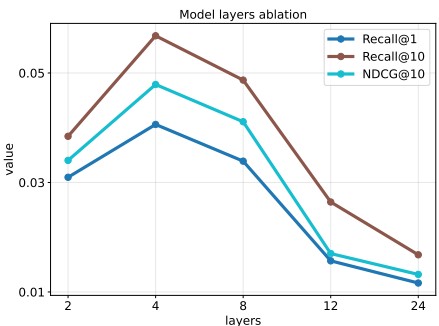
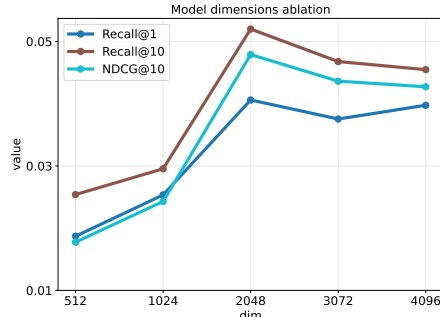

(a) **Ablation study of model dimensions.**

(b) **Ablation study of number layers.**

Figure 3: **Results of ablation experiments for model dimensions and number of layers**.

### 4.3 ABLATION STUDY

In this section, we set up multiple ablation study to verify the validity of our proposed approach. We first measure the superiority of our proposed quantization named `UIQ`. The results are shown in Table 2. We see that `UIQ` achieves SOTA compared with other methods and the metrics improve obviously. The quantization method of assigning random ids achieved the lowest performance, which also confirmed the importance of the *Preservation of Semantic Topology* we mentioned for generative recommendations. Compared with RQ methods (*i.e.* RQ-VAE and RQ-Kmeans), our proposed approach further enhances performance because our approach attempts to enable the model to identify some mutually independent decisive features from a global perspective rather than forcibly introducing a chain hierarchy. Ablation experiments on $\beta$ and $\gamma$ during training are presented in Appendix F.1. The results of ablation experiments with respect to model parameter Settings are shown in Figure 3. More ablation study results can be found in Appendix F.

Table 2: Abalation study of various discrete quantization schemes.

| Methods | Arts Crafts and Sewing | | | | |
| --- | --- | --- | --- | --- | --- |
| | Recall@1 | Recall@5 | NDCG@5 | Recall@10 | NDCG@10 |
| Random ID | 0.0146 | 0.0147 | 0.0124 | 0.0153 | 0.0136 |
| RQ-VAE | 0.0321 | 0.0407 | 0.0381 | 0.0348 | 0.0292 |
| RQ-Kmeans | 0.0312 | 0.0449 | 0.0371 | 0.0483 | 0.0395 |
| `UIQ` | **0.0410** | **0.0520** | **0.0464** | **0.0568** | **0.0479** |

## 5 CONCLUSION

In this paper, we propose a new generative recommendation paradigm named `GREED`, which transforms item generation to a multi-token pattern and increases the planning ability of model. We first use a novel uniform implicit quantization scheme instead of VQ and RQ to improve the quantity of discrete representation of items. And we design a new multi-token discrete diffusion architecture rather than ARM to avoid imposing causal relationships on items and alleviate the problem of error accumulation caused by error item generation. Our proposed `GREED` achieves the SOTA performance compared to all discriminative and generative recommendation methods. And `GREED` has higher generation efficiency and more scalable application capabilities for different recommendation tasks. We believe this provides a valuable and significant research approach for future GRs.

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

# A RELATED WORK

## A.1 RECOMMENDATION SYSTEMS

**Sequential Recommendation** Approaching sequential recommendation as a time-series prediction task, several methods have employed recurrent architectures such as GRUs to model user behavior sequences (Hidasi et al., 2015; Li et al., 2017). Tang & Wang (2018) represents user interaction sequences as images in both time and latent spaces, employing convolutional filters to extract local features that effectively capture sequential patterns and skip behaviors. With the rise of attention mechanisms, subsequent work incorporated self-attention to better capture item dependencies and user preferences (Zhang et al., 2018; Kang & McAuley, 2018; Li et al., 2021; Zhang et al., 2019; Klenitskiy & Vasilev, 2023). The success of the Transformer architecture in sequence modeling further motivated its adaptation for recommendation tasks. For instance, Sun et al. (2019) and de Souza Pereira Moreira et al. (2021) developed dedicated Transformer-based models for sequential recommendation. More recently, Zhou et al. (2020) proposed $S^3$-Rec, which enhances item representations through self-supervised pre-training based on mutual information maximization (MIM), aiming to capture multi-granular correlations in item sequences and improve recommendation performance.

**Generative Recommendation** Generative models have exhibited remarkable generalization capabilities, with studies showing that they develop emergent abilities as data and model size increase. Recently, generative recommendation has attracted growing research interest. Geng et al. (2022) introduced P5, which unifies multiple recommendation tasks into a text generation framework using personalized prompts, enabling knowledge sharing and zero-shot recommendation capabilities through a pre-trained Transformer architecture. Similarly, Zhai et al. (2024) incorporated discrete features into a unified long-term interaction sequence, scaling the model to hundreds of trillions of parameters and exploring scaling laws for large generative recommendation systems. To reduce the parameter footprint, alternative methods represent items as discrete token sequences via quantization techniques, then train generative models to produce these discrete sequences as targets (Rajput et al., 2023; Deng et al., 2025).

## A.2 DISCRETE DIFFUSION MODELS

Diffusion models have gained significant traction due to their strong generative performance and output diversity. Recent work has begun exploring their application to discrete data domains. Austin et al. (2021) proposed D3PM, which incorporates structured transition matrices to model discrete corruption processes and introduces an auxiliary loss to form a generalized discrete diffusion probabilistic model. Building on this, Li et al. (2022) introduced Diffusion-LM, which generates text by iteratively denoising Gaussian noise into word embeddings and leverages continuous intermediate latent variables for gradient-based controllable text generation. Gong et al. (2023) presented DiffuSeq, a sequence-to-sequence text generation framework based on diffusion models that performs partial noising and conditional denoising in continuous latent space to produce high-quality and diverse outputs. Another line of work gradually replaces discrete tokens (e.g., text or image tokens) with mask tokens and trains a model to reconstruct the original data by reversing this process (Shi et al., 2024; Sahoo et al., 2024). These advances have laid the theoretical and practical groundwork for large-scale discrete diffusion language models, as demonstrated in recent studies such as Nie et al. (2025) and Ye et al. (2025).

# B THEOREM PROVING

## B.1 THE VALIDITY OF `UIQ`

Our `UIQ` can achieve the Rate-Distortion Trade-off during the whole quantization process. We know that the scalar quantization can lead distortion because of round operation. By regarding finite scalar quantization as a uniform scalar quantizer, we can obtain the following theorem.

**Corollary 1** (**Optimality of Uniform Input Distribution for Scalar Quantization**) *Let* $\mathbf{z} \in \mathbb{R}^d$ *be a continuous latent representation, and let* $\mathbf{q} = \mathbf{round}\left[f(\mathbf{z})\right] \in \{0, 1, \ldots, L-1\}^d$ *denote its*

*finite scalar quantization with $L$ levels per dimension. If the input $\mathbf{z}$ follows a standard uniform distribution with independent dimensions, i.e., $\mathbf{z} \sim U(0,1)^d$, then:*

- The expected quantization distortion $\mathbb{E}[\mathcal{D}_{\text{quant}}]$ is minimized, which ensures maximal preservation of original information in the discrete representation.

- The entropy of the discrete code $\mathbf{q}$ is maximized, reaching the upper bound of $d \cdot \log L$, which guarantees optimal codebook utilization and expressive power.

*Proof*: The result follows directly from Theorems 1 and 2.

**Theorem 1** For a uniform scalar quantizer given a level number $L$, if the input $z$ follows a standard uniform distribution $U(0,1)$, then the expected distortion $\mathbb{E}[\mathcal{D}_{\text{quant}}]$ is minimized.

*Proof*:

We first recall Lloyd's conditions from rate-distortion theory, which provide necessary criteria for optimal quantizer design.

**Definition 1 (Lloyd's Conditions)** Let $X$ be a random variable quantized into $L$ distinct levels $\{y_1, y_2, \ldots, y_L\}$ with partition boundaries $\{b_0, b_1, \ldots, b_L\}$ where $b_0 = -\infty$ and $b_n = \infty$, The quantizer is optimal if:

- **Minimum Distortion Reconstruction Condition** Each quantization level $y_i$ is defined as the conditional expectation of the input values within its corresponding quantization interval:

$$y_i = \mathbb{E}[X \mid X \in [b_{i-1}, b_i)] \tag{13}$$

- **Optimal Partition Condition** Each boundary $b_i$ of the quantization intervals is defined as the midpoint between adjacent quantization levels:

$$b_i = \frac{y_i + y_{i+1}}{2} \tag{14}$$

Let $\mathbf{z}$ represents the vector to be quantized, and $Q(\cdot)$ reprsent quantization operation, we represent distortion $\mathcal{D}_{\text{quant}}$ by using the mean square error of the values before and after quantization as follows.

$$\mathcal{D}_{\text{quant}} = \mathbb{E}[(z - \tilde{z})^2] \tag{15}$$

where $\tilde{z} = Q^{-1}(q)$, $Q^{-1}(\cdot)$ is the anti-quantization operation. For a uniform quantizer, when the probability density function (PDF) of the input variable $Z$ is $p(z)$, its expected distortion is:

$$\mathbb{E}[\mathcal{D}_{\text{quant}}] = \int_{-\infty}^{+\infty} \left[z - \tilde{z}\right]^2 p(z) dz \tag{16}$$

$$= \int_0^1 \left[z - Q^{-1}(Q(z))\right]^2 p(z) dz \tag{17}$$

$$= \sum_{k=0}^{L-1} \int_{I_k} \left[z - Q^{-1}(Q(z))\right]^2 p(z) dz \tag{18}$$

$$= \sum_{k=0}^{L-1} \int_{\frac{k}{L}}^{\frac{k+1}{L}} \left[z - Q^{-1}(Q(z))\right]^2 p(z) dz \tag{19}$$

When input distribution $p(z)$ is uniform distribuiton $z \sim U(0,1)$, $p(z) = 1$ for $z \in [0,1]$. Considering uniform quantizer has $L$ quantization levels, $I_k$ is $[\frac{k}{L}, \frac{k+1}{L})$ $k = 0, 1, \ldots L - 1$, and reconstruct

value $q_k = \frac{2k+1}{2L}$. For each interval, the centroid is as follows.

$$\mathbb{E}[z \mid z \in I_k] = \frac{\int_{\frac{k}{L-1}}^{\frac{k+1}{L}} z \cdot 1 dz}{\int_{\frac{k}{L}}^{\frac{k+1}{L}} 1 dz} \tag{20}$$

$$= \frac{\frac{1}{2} \left[ \left( \frac{k+1}{L} \right)^2 - \left( \frac{k}{L} \right)^2 \right]}{\frac{1}{L}} \tag{21}$$

$$= \frac{2k+1}{2L} = q_k \tag{22}$$

which meets Minimum Distortion Reconstruction Condition. And the midpoint of reconstruct value $q_k$ and $q_{k+1}$ is

$$\frac{q_k + q_{k+1}}{2} = \frac{\frac{2k+1}{2L} + \frac{2k+3}{2L}}{2} = \frac{4k+4}{4L} = \frac{k+1}{L} \tag{23}$$

which is the bound of interval $I_k$ and $I_{k+1}$. So it meets the Optimal Partition Condition. Because the uniform quantizer satisfies Lloyd's Conditions, it is optimal for uniform distribution $U(0,1)$, that is, expected distortion $\mathbb{E}[\mathcal{D}_{\text{quant}}]$ is minimized.

Let $u = z - \frac{2k+1}{2L}$, then $u = -\frac{1}{2L}$ when $z = \frac{k}{L}$. $u = \frac{1}{2L}$ when $z = \frac{k+1}{L}$. So we can obtain

$$\mathbb{E}[\mathcal{D}_{\text{quant}}] = \sum_{k=0}^{L-1} \int_{\frac{k}{L}}^{\frac{k+1}{L}} \left[ z - Q^{-1}(Q(z)) \right]^2 p(z) dz \tag{24}$$

$$= \sum_{k=0}^{L-1} \int_{\frac{k}{L}}^{\frac{k+1}{L}} (z - \frac{2k+1}{L})^2 dz \tag{25}$$

$$= L \int_{-1\frac{1}{2L}}^{\frac{1}{2L}} u^2 du \tag{26}$$

$$= L * \frac{1}{12L^3} \tag{27}$$

$$= \frac{1}{12L^2} \tag{28}$$

Therefore, under a uniform distribution, the uniform quantizer can achieve the minimum expected distortion $1/12L^2$.

The working objective of the VQ series is to increase entropy as much as possible to maximize the utilization of codebooks. This means encouraging the model to fully utilize all the codewords in the codebook to capture the diverse semantic information of the input data and avoid low codebook utilization. Here we present **Theorem 2** for higher utilization of `UIQ` codebooks.

**Theorem 2** Under the premise that each dimension is independent, when each discrete variable $Q_i$ is uniformly distributed, that is, $P(Q_i = k) = 1/L$ for all $k \in \{0, 1, ..., L-1\}$, the entropy $H(\mathbf{Q})$ reaches its maximum value of $d \cdot \log L$.

*Proof*:

**Definition 2 (Jensen's Inequality)** Let $\varphi : I \rightarrow \mathbb{R}$ be a convex function, where $I \subseteq \mathbb{R}$ is an interval. For any random variable $X$ that satisfies the following conditions:

- The value of $X$ is within $I$ (that is, $X \in I$ almost necessarily holds)
- The expectations of $X$ and $\varphi(X)$ exist

Then Jensen's inequality holds as follows.

$$\varphi\left(\mathbb{E}[X]\right) \leq \mathbb{E}\left[\varphi(X)\right] \tag{29}$$

If $\varphi$ is a concave function, then the direction of the inequality is opposite:

$$\varphi\left(\mathbb{E}[X]\right) \geq \mathbb{E}\left[\varphi(X)\right] \tag{30}$$

*Discrete Version* For convex functions $\varphi$, if $\lambda_1, \lambda_2, \ldots, \lambda_n \geq 0$ and $\sum_{i=1}^{n} \lambda_i = 1$, then:

$$\varphi\left(\sum_{i=1}^{n} \lambda_i x_i\right) \leq \sum_{i=1}^{n} \lambda_i \varphi(x_i) \tag{31}$$

For discrete random variables $Q_i$, the probability mass function is $p(q) = \mathbb{P}(Q_i = q)$, which satisfies $\sum_{q=0}^{L-1} p(q) = 1$ and $p(q) \geq 0$. Entropy is defined as

$$H(Q_i) = -\sum_{q=0}^{L-1} p(q) \log p(q) = \mathbb{E}\left[\log \frac{1}{p(Q_i)}\right] \tag{32}$$

Here, the expectation is the distribution of $Q_i$.

By Jensen's Inequality, for the concave function $\varphi$ and the random variable $X$, we have

$$\mathbb{E}[\varphi(X)] \leq \varphi(\mathbb{E}[X]) \tag{33}$$

And the equal sign holds if and only if $X$ is a constant almost everywhere (that is $X$ is constant with a probability of 1.

Here, let $X = \frac{1}{p(Q_i)}$, then $\log X = \log \frac{1}{p(Q_i)}$ is a concave function (because $\log$ is a concave function).

$$\mathbb{E}\left[\log \frac{1}{p(Q_i)}\right] \leq \log \mathbb{E}\left[\frac{1}{p(Q_i)}\right] \tag{34}$$

That is,

$$H(Q_i) \leq \log \mathbb{E}\left[\frac{1}{p(Q_i)}\right] \tag{35}$$

$$= \log \sum_{q=0}^{L-1} p(q) \cdot \frac{1}{p(q)} \tag{36}$$

$$= \log \sum_{q=0}^{L-1} 1 = \log L \tag{37}$$

So,

$$H(Q_i) \leq \log L \tag{38}$$

The equality holds if and only if the equality in Jensen's Inequality holds, that is, $p(Q_i)$ is a constant almost everywhere. This means there exists a constant $c$ such that for all $q$ in the support set of $Q_i$ (i.e., $p(q) > 0$), $1/p(q) = c$, that is, $p(q) = 1/c$.

Because the probability mass function satisfies $\sum_{q=0}^{L-1} p(q) = 1$, substituting it yields

$$\sum_{q=0}^{L-1} \frac{1}{c} = L \cdot \frac{1}{c} = 1 \tag{39}$$

Solving for $c$ gives $c = L$. Therefore, for all $q$, there is $p(q) = \frac{1}{L}$, which means that $Q$ follows a uniform distribution. Conversely, if $Q$ follows a uniform distribution, that is, for all $q$, $p(q) = \frac{1}{L}$, then

$$H(Q_i) = -\sum_{q=0}^{L-1} \frac{1}{L} \log \frac{1}{L} = \log L \tag{40}$$

Therefore, the equal sign holds. In conclusion, the maximum value of entropy $H(Q_i)$ is $\log L$, which is obtained if and only if $Q_i$ follows a uniform distribution. Therefore, when all $Q_i$ are uniformly distributed, the total entropy is the maximum

$$\max H(\mathbf{Q}) = d \cdot \log L \tag{41}$$

## C  DATASET DETAILS

We evaluated the proposed method on three public benchmarks constructed from the Amazon Product Reviews dataset (2018) (He & McAuley, 2016): Arts, Crafts and Sewing; Video Games; and Musical Instruments. The dataset contains user reviews of Amazon products spanning from May 1996 to October 2018. Table 3 summarizes the detailed statistics. User interaction sequences were built by chronologically sorting all reviews per user, and users with fewer than 5 interactions were filtered out. In line with standard evaluation protocols in sequential recommendation (Kang & McAuley, 2018; Zhou et al., 2020; Rajput et al., 2023), we employed a leave-one-out strategy: for each user's interaction sequence, the most recent item was held out for testing, the second most recent for validation, and all earlier interactions were used for training.

Table 3: The statistic details of dataset.

| Dataset | User | Item | Rating | Mean $l$ | Med $l$ | Sparsity |
|---|---|---|---|---|---|---|
| Arts Crafts and Sewing | 56,193 | 22,931 | 490,853 | 8.735 | 7.00 | 99.96% |
| Video Games | 55,220 | 17,408 | 495,728 | 8.97 | 6.00 | 99.96% |
| Musical Instruments | 27,520 | 10620 | 230,319 | 8.369 | 6.00 | 99.93% |

## D  BASELINE DETAILS

In this section, we briefly introduce the details of the different baseline methods we compared.

- GRU4Rec (Hidasi et al., 2015) proposes a recurrent neural network (RNN) approach for session-based recommendations, which significantly improves accuracy by modeling the entire user session and incorporating tailored modifications like a ranking loss function, addressing the limitations of traditional methods in scenarios with short user histories.

- Bert4Rec (Sun et al., 2019) models user behavior sequences by adopting a deep bidirectional self-attention network based on Transformer and combining it with the Cloze task (i.e., predicting randomly masked items in the sequence) to achieve more accurate sequence recommendations.

- HGN (Ma et al., 2019) proposed a Hierarchical Gating Network model to address the challenge of modeling long-term and short-term interests of users in sequential recommender systems. Through feature gating, instance gating and item-item product module, HGN adaptively selects important features and items, and explicitly captures the relationship between items.

- SASRec (Kang & McAuley, 2018) combines graph convolutional networks to capture user-item relationships and self-attention sequence models for prediction, and enhances representation capabilities in a multi-task learning framework through instance-level and prototype-level contrastive learning.

- S³-Rec (Zhou et al., 2020) integrates the Mutual Information Maximization (MIM) principle into the self-attention recommendation architecture and designs four self-supervised learning objectives to capture the intrinsic correlations among items, attributes, subsequences, and sequences, thereby enhancing data representation and improving sequence recommendation performance during the pre-training stage.

- P5 (Geng et al., 2022) uniformly converts all recommendation-related data such as user-product interaction, metadata and comments into natural language sequences, designs personalized prompt templates, and uses the pre-trained Encoder-Decoder Transformer model to solve up to five types of recommendation tasks in a text generation manner. So as to achieve knowledge sharing, zero-shot generalization ability and a unified recommendation paradigm.

- Tiger (Rajput et al., 2023) proposed a generative retrieval recommendation system framework, which represents items as a series of Semantic ids generated by quantifying content features through RQ-VAE. And use the Transformer-based sequence-to-sequence model to autoregressively predict the Semantic ID of the item that the user will interact with next.

- OneRec (Deng et al., 2025) proposes a unified generative encoder-decoder model, which utilizes sparse MoE to expand model capacity, recommends video lists through sstor-level generation rather than point-by-point prediction, and continuously optimizes recommendation quality by combining an iterative preference alignment mechanism with a reward model and a custom sampling strategy.

# E  IMPLEMENTATION DETAILS

To obtain the representations of items, we employ pre-trained Sentence-T5 model (Raffel et al., 2020) to encode the items. For the input features, we use item's content features like title, category, price and brand followed (Rajput et al., 2023). Through leveraging these features to constitue sentences, we can obtain the item's semantic representations of 768 dimension.

The quantization module employs a Multi-Layer Perceptron (MLP) architecture for both the encoder and decoder, using ReLU activation functions. Each MLP consists of three intermediate layers with dimensions 512, 256, and 128, respectively. The quantization level is set to $L = 51$, and for fair comparison with Tiger and Onerec, the number of codebook layers $d$ is set to 3. Following the practice in Tiger, items that collide under the same semantic ID are disambiguated by flattening them using a fourth positional token. We adopt the T-NAF flow model (Patacchiola et al., 2024) to enhance expressivity. To ensure training stability, the encoder and decoder are first trained independently, allowing the encoder to learn semantically meaningful representations. After this initial phase, the encoder parameters are frozen, and training continues for the flow model and decoder. The entire quantization model is optimized using the AdamW optimizer with a learning rate of 0.001 and a batch size of 2048. In the loss function, we set the weighting factor $\lambda = 0.5$ to balance the optimization of both the flow model and the reconstruction task performed by the decoder.

For our generative recommendation model based on a diffusion language model architecture, we adopt the framework established in (Nie et al., 2025; Ye et al., 2025). To enable the model to handle sequential recommendation tasks, we design the vocabulary of the sequence-to-sequence model to include tokens corresponding to each semantic ID. The theoretical vocabulary size is $51 \times 4 = 204$ tokens; however, the actual utilized vocabulary size $|\mathcal{V}| < 204$ due to the unique design of our UIQ method, which significantly reduces token collision. The diffusion language model is configured with 4 layers and a hidden dimension of 2048. We employ RMSNorm (Zhang & Sennrich, 2019) as the normalization method with $\epsilon = 1 \times 10^{-5}$, and use the SiLU activation function. During training, we set $\beta = 0.5$ and $\gamma = 0.5$. The model is optimized using AdamW with a learning rate of 0.0005 and a weight decay of 0.01.

All my experiments were done using the Nvidia H100 with Linux system.

# F  MORE ABLATION STUDY

## F.1  ABLATION STUDY OF TRAINING STAGE

The ablation results exploring the impact of hyperparameters $\beta$ and $\gamma$, which control the range of the uniform distribution used during training, are shown in Figure 4. The results indicate that the values of $\beta$ and $\gamma$ significantly influence model performance. This is due to the fact that tokens are masked to the [**MASK**] state with probability $1 - \alpha_t$. By adjusting $1 - \alpha_t$ to follow a uniform distribution $U[\gamma, \beta + \gamma]$ through $\beta$ and $\gamma$, the model can adaptively focus on denoising tasks of suitable difficulty during training (*e.g.* When the $1 - \alpha_t$ is set too low, the model faces an overly simple reconstruction task. This simplicity yields weak learning signals, leading to degraded optimization performance). This mechanism helps mitigate excessive gradient variance, leading to more stable and efficient optimization.

## F.2  COLLISION

We present a visual analysis of the quantization collision rates for our proposed UIQ and existing methods, by examining the most frequently occurring semantic ID sequences (Figure 5). The results indicate that RQ-VAE continues to suffer from codebook collapse, crowding a large number of items into the same semantic ID sequence. This not only limits codebook utilization but also disrupts

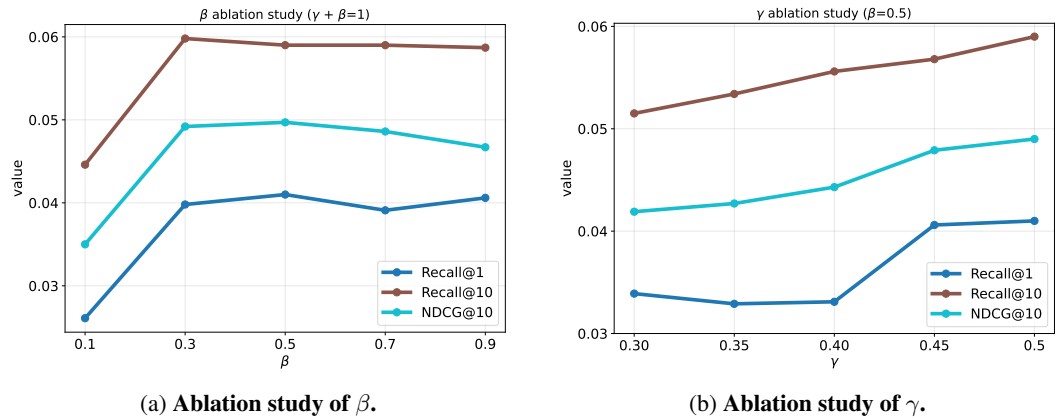

(a) **Ablation study of $\beta$.**  (b) **Ablation study of $\gamma$.**

Figure 4: **Results of ablation experiments for $\beta$ and $\gamma$.**

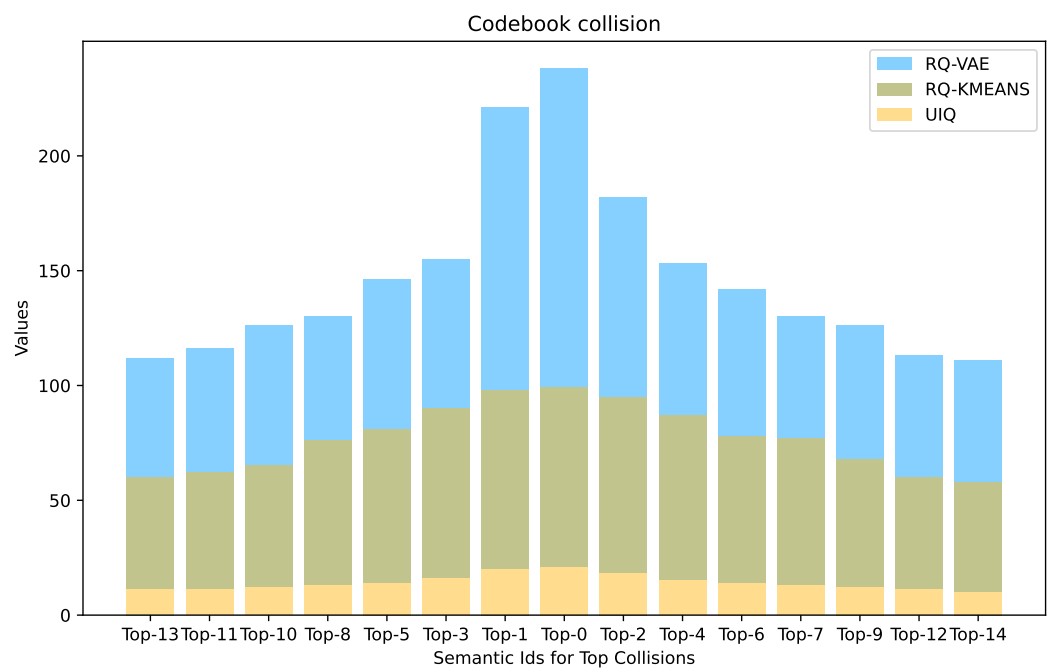

Figure 5: **Statistics of the number of items under semantic ids with the highest collision rate of codebook.**

the relative semantic relationships between items. In contrast, our approach significantly reduces the collision rate, improves codebook utilization, and better preserves inter-item semantics. As a result, it alleviates the burden on the generative model during inference and enhances the overall performance of generative recommendation.

### F.3 NEW ABILITIES

**Conditional Item Recommendation** Leveraging the inherent non-autoregressive generation capability of diffusion language models, our GREED paradigm is well-suited for conditional recommendation tasks. As demonstrated in Table 4, GREED can faithfully generate recommendations when provided with any number and any order of semantic ID tokens as preconditions (denoted as $n$-token conditions). This flexibility directly addresses a key real-world scenario: even when users only provide partial or loosely ordered conditions, GREED can still deliver accurate recommendations.

Table 4: New conditional generation capability.

| Methods | Arts Crafts and Sewing | | |
|---|---|---|---|
| | Recall@1 | Recall@5 | NDCG@5 |
| 1-token conditions | 0.16 | 0.45 | 0.32 |
| 2-token conditions | 0.43 | 0.62 | 0.43 |
| 3-token conditions | 0.52 | 0.73 | 0.55 |
| w/o token conditions | 0.041 | 0.052 | 0.046 |

**Trade Off between Performance and Efficiency** By adjusting the top-k sampling size, our `GREED` framework offers a flexible trade-off between recommendation performance and computational efficiency. More importantly, for latency-critical scenarios, `GREED` can generate the entire semantic ID sequence in a single parallel step, unlike the sequential token-by-token generation of autoregressive (ARM) models. This non-autoregressive capability is a significant advantage for real-world recommendation systems where low inference latency is critical.

## G  DISCUSSION

Our `GREED` model is the first to utilize an independent spatial semantic ID and a diffusion language model to model user historical behavior sequences. Compared to autoregressive models, our approach models multi-dimensional semantics of items in an independent discrete semantic space, avoiding the issue of semantic ID contextual modeling failure caused by error accumulation. In terms of diversity, each denoising step starts with the most confident token for beam search, generating candidate sets based on cumulative probability scores, thereby ensuring diversity in generation. In future work, we can explore more optimal diversity generation strategies and employ sampling methods that better align with the characteristics of `UIQ` to achieve even more outstanding performance.

## H  LLM USAGE

We utilized large language models (*e.g.* GPT-4o and Gemini) exclusively for auxiliary tasks related to text polishing and document formatting. This assistance was limited to grammatical corrections, phrasing improvements, and suggestions on the presentation of figures and tables. The LLMs played no role in the core research processes, including the generation of ideas, design of experiments, implementation, data analysis, or development of technical content. The authors carefully reviewed and edited all AI-assisted output and bear complete responsibility for the final manuscript.

