# OpenReview forum: "Diffusion Beats ARM: Diffusion Large Language Models for Generative Recommendation"
_ICLR.cc/2026/Conference — ICLR 2026 Conference Withdrawn Submission_

### Official Review · Reviewer_UQhQ · 2025-10-18

**Soundness:** 3
**Presentation:** 3
**Contribution:** 3
**Rating:** 4
**Confidence:** 4

**Summary:**

This paper proposes a diffusion language model paradigm for generative recommendation, instead to autoregressive models (ARMs). It argues that ARMs are fundamentally misaligned with recommendation tasks because item attributes are parallel and interdependent rather than sequential. To address this, it proposes two main contributions: 1. Uniform implicit quantization (UIQ) that uses normalizing flows to transform item embeddings into uniform distributions before scalar quantization, avoiding codebook collapse. 2. A hybrid generation architecture that uses discrete diffusion to generate individual items while using autoregression across items in the sequence.

**Strengths:**

1. This paper proposes uniform implicit quantization (UIQ), addressing codebook collapse in VQ-based methods and artificial sequential dependencies in RQ-based methods.
2. The proposed GREED achieves strong performance compared to both discriminative methods and generative methods.
3. It is worth exploring using diffusion language models on recommendation tasks, as long as there are clear motivations.

**Weaknesses:**

1. The paper claims that "diffusion beats ARM", but the evidence suggests that UIQ is the core contribution to improve the performance. If there is a well designed quantization for the ARM (e.g., TIGIER, OneRec), then the ARM may still be a better choice than diffusion language models for recommendation.
2. The motivation of using diffusion language models claims item attributes are parallel. Tthis motivation is farfetched. And the ablation results show that improvements are mainly from UIQ. Why parallel generation of semantic IDs beat sequential generation is confusing. The paper provides intuition but no evidence.
3. All datasets are from Amazon Review dataset. It limits the domain diversity. The average sequence length in Amazon Review dataset is short (< 10).

**Questions:**

Please address the weaknesses above.

1. The paper makes a valuable contribution (UIQ) but oversells another contribution (diffusion) without sufficient evidence. The title and framing may be misleading given the actual results. And is it possible to use UIQ + ARM? I would like to improve my score if the authors clearly demonstrate or explain the benefits of diffusion language models or reframing as primarily a quantization contribution with diffusion as a reasonable approach.

2. How does GREED perform compared with diffusion-based sequential recommendation models? It would be better to cite some of these methods [1][2][3] and acknowledge their contributions.

3. It would be more clear if the authors can provide the source code.

[1] DiffuRec: A Diffusion Model for Sequential Recommendation. Li et al.

[2] Generate What You Prefer: Reshaping Sequential Recommendation via Guided Diffusion. Yang et al.

[3] A Survey on Diffusion Models for Recommender Systems. Lin et al.

---

### Official Review · Reviewer_ddH7 · 2025-10-25

**Soundness:** 3
**Presentation:** 2
**Contribution:** 2
**Rating:** 2
**Confidence:** 4

**Summary:**

The paper proposes GREED (Generative REcommendation via Elemental Diffusion over Large Language Models), a novel generative recommendation system that leverages diffusion-based modeling to address limitations in autoregressive models (ARMs) for discrete quantized generative recommendation (DQGR). It argues that ARMs impose artificial sequential dependencies on parallel item attributes, leading to errors and inefficiency. GREED introduces Uniform Implicit Quantization (UIQ) to enhance discrete item representation and a diffusion-based paradigm for parallel multi-token prediction. Experiments on benchmark datasets show GREED outperforming state-of-the-art methods, with claims of improved efficiency and ranking performance, though validation details are sparse.

**Strengths:**

- **Originality**: Attempts to apply diffusion to DQGR, a departure from ARM dominance, though heavily derivative.
- **Quality**: Some performance gains are reported, but experimental design flaws undermine reliability.
- **Clarity**: Limited by poor figure annotation and lack of detailed methodology.
- **Significance**: Targets efficiency, but lacks evidence of real-world impact or scalability.

**Weaknesses:**

- **Methodological Flaws**: UIQ’s codebook utilization claim lacks empirical validation (e.g., no entropy or diversity metrics). The diffusion process’s parallel advantage is not quantified across varying sequence lengths or compared to optimized ARMs (e.g., with parallel decoding).
- **Experimental Gaps**: No statistical analysis (e.g., t-tests) supports performance claims. Dataset specifics (e.g., item count, attribute diversity) and hyperparameter tuning are omitted, hindering reproducibility.
- **Oversight**: Ignores potential biases in quantized representations and their impact on user fairness. Efficiency claims lack comparison with memory-optimized diffusion models (e.g., DDIM).
- **Validation**: Offline analyses are mentioned without metrics or protocols, and no online testing data is provided to substantiate real-world applicability.

**Questions:**

1. Can the authors provide statistical tests (e.g., t-tests) to validate UIQ’s superiority over VQ and RQ in codebook utilization and semantic fidelity?
2. How does GREED’s performance scale with increasing sequence lengths, and why was no comparison made with parallel-decoding ARMs (e.g., Transformer-XL)?
3. What were the dataset sizes, attribute distributions, and hyperparameter settings used in experiments, and how were they tuned to ensure robustness?
4. Can the authors quantify the error propagation in ARMs versus diffusion in GREED with a controlled experiment on synthetic data?
5. Why were no memory-optimized diffusion variants (e.g., DDIM) compared to assess GREED’s efficiency claims?

---

### Official Review · Reviewer_e48C · 2025-10-31

**Soundness:** 2
**Presentation:** 3
**Contribution:** 2
**Rating:** 4
**Confidence:** 2

**Summary:**

The paper proposes a generative recommendation framework named GREED based on diffusion large models. It represents items as discrete semantic IDs through uniform implicit quantization (UIQ), and replaces the traditional ARM with non-autoregressive block-level diffusion generation to alleviate the feature sequence assumption and error accumulation. The experiments have achieved state-of-the-art results on three Amazon datasets.

**Strengths:**

1.For the first time, discrete diffusion was applied to recommendation, aligning with the parallel structure of item attributes and improving generation efficiency.

2.UIQ utilizes learnable flows to explicitly align uniform distributions, theoretically proving to minimize quantization distortion and maximize codebook entropy, overcoming VQ collapse.

3.Block-level multi-token parallel generation supports global planning, with an average Recall@1 improvement of over 15%, and allows for flexible adjustment of sampling size to balance performance and latency.

4.New capabilities such as conditional recommendation and list sorting are provided, verifying the scalability of the framework.

**Weaknesses:**

1. When compared with strong baselines such as S3-Rec, the pre-training data sizes were not unified. Using Sentence-T5 encoding in GREED might introduce additional external semantics, raising doubts about the fairness.

2.UIQ relies on the learnable flow to fit the edge distribution. When dealing with high-dimensional sparse features and having insufficient training samples, the flow model is prone to overfitting, resulting in an increase in quantization error. The paper does not provide high-dimensional experiments or regularization strategies.

3.Discrete diffusion requires maintaining a large-sized transfer matrix, and the video memory grows linearly with the number of codebook layers. When the scale of the product reaches the tens of millions level, distributed training or approximate sampling schemes have not been discussed. The engineering feasibility needs to be verified.

**Questions:**

see weakness

---

### Official Review · Reviewer_Veks · 2025-11-10

**Soundness:** 2
**Presentation:** 1
**Contribution:** 1
**Rating:** 2
**Confidence:** 4

**Summary:**

This paper addresses the misalignment between autoregressive models (ARMs) and the parallel, sequence-free nature of items in generative recommendation, proposing the GREED paradigm. It has two stages: 1) Uniform Implicit Quantization; 2) discrete diffusion generation (non-autoregressive parallel generation to reduce errors, block-level autoregression for list-wise recommendations). Experimental results on three public Amazon datasets validate the effectiveness of GREED.

**Strengths:**

* S1: This paper proposes a non-autoregressive discrete diffusion generation paradigm, GREED, that breaks free from the sequential dependencies imposed by traditional autoregressive models.
* S2: Through probability integral transformation (PIT) and normalized flow, item representations are transformed into uniform distributions, theoretically demonstrating minimal quantization distortion and maximum codebook entropy.
* S3: GREED supports flexible conditional recommendation tasks, capable of handling arbitrary numbers and sequences of attribute condition inputs to accommodate users' ambiguous needs in real-world scenarios.

**Weaknesses:**

* W1: The dataset scenario is limited, and generalization validation is insufficient. Validation was conducted solely on three product categories from Amazon Product Reviews (Arts, Video Games, and Musical Instruments), all of which fall within e-commerce product recommendation scenarios. This fails to demonstrate the method's effectiveness in scenarios with high cardinality and more frequent dynamic updates.
* W2: The comparison baseline is outdated, and many of the latest generative recommendation baselines were not included in the comparison. The experimental results lack credibility.
* W3: The comparison and ablation experiments for core components are insufficiently comprehensive. UIQ is only contrasted with RQ-VAE and RQ-Kmeans, omitting newer quantization methods (e.g., adaptive quantization, hybrid quantization). The impact of quantization level L and stream model type on performance was not analyzed.

**Questions:**

* Q1: You need to supplement the latest baseline methods, such as HSTU,  LCRec, MiniOneRec, and MTGR.
* Q2: The impact of quantization level L and stream model type on performance should be analyzed.
* Q3: Testing with a single dataset alone cannot demonstrate the method's universality; additional datasets are required to validate its generalization capability (provided that the latest baseline is used for comparison).

---

### Note · Authors · 2025-11-27

I have read and agree with the venue's withdrawal policy on behalf of myself and my co-authors.